# Upgrading VAE Training With Unlimited Data Plans Provided by Diffusion Models

## Abstract

Variational autoencoders (VAEs) are popular models for representation learning but their encoders are susceptible to overfitting (Cremer et al., 2018) because they are trained on a finite training set instead of the true (continuous) data distribution $p_{\text{data}}(\boldsymbol{x})$. Diffusion models, on the other hand, avoid this issue by keeping the encoder fixed. This makes their representations less interpretable, but it simplifies training, enabling accurate and continuous approximations of $p_{\text{data}}(\boldsymbol{x})$. In this paper, we show that overfitting encoders in VAEs can be effectively mitigated by training on samples from a pre-trained diffusion model. These results are somewhat unexpected as recent findings (Alemohammad et al., 2023; Shumailov et al., 2023) observe a decay in generative performance when models are trained on data generated by another generative model. We analyze generalization performance, amortization gap, and robustness of VAEs trained with our proposed method on three different data sets. We find improvements in all metrics compared to both normal training and conventional data augmentation methods, and we show that a modest amount of samples from the diffusion model suffices to obtain these gains.

## 1 Introduction

Variational autoencoders (VAEs, Kingma & Welling (2014); Rezende et al. (2014)) are a class of deep probabilistic models. They model the underlying data distribution $p_{\text{data}}(\boldsymbol{x})$ from which a given training set $\mathcal{D}_{\text{train}} = \{\boldsymbol{x}_i\}_{i=1}^{N}$ was drawn. Beyond their generative modeling capabilities, VAEs have many other favorable properties by design which lead to applications such as representation learning (van den Oord et al., 2017) and compression (Yang et al., 2023). However, these properties can be compromised if the VAE is overfitted. Specifically, the encoder $f_\phi(\boldsymbol{x})$ is more susceptible to overfitting (Wu et al., 2017; Cremer et al., 2018; Shu et al., 2018) than the decoder since a finite training set $\mathcal{D}_{\text{train}}$ is repeatedly fed into the encoder. By contrast, the decoder is trained on unique samples from the approximate posterior distribution inferred by the encoder.

Overfitting in the encoder implies that the learned mapping $f_\phi(\boldsymbol{x})$ does not generalize well to unseen data, which can negatively impact the performance of generative modeling, amortized inference, and adversarial robustness. For generative modeling, as the number of training epochs increases, an overfitted VAE will have a higher evidence lower bound (ELBO) on the training set but a lower ELBO on the test set. For amortized inference, an overfitted encoder is more likely to map unseen data to a suboptimal set of variational parameters. This results in a lower ELBO when compared to the ELBO obtained by directly optimizing these parameters. For robustness, an overfitted encoder often learns a less smooth $f_\phi(\boldsymbol{x})$, such that a small change in the input $\boldsymbol{x}$ can result in a large difference in the latent space. This makes VAEs vulnerable to adversarial attacks, causing realistic and hardly distinguishable inputs to yield semantically different outputs (Kuzina et al., 2022).

One major cause of overfitting in VAEs is the multiple iterations over the insufficient amount of training data (more details in Section 5). Ideally, we aim to train VAEs with unique samples drawn from $p_{\text{data}}(\boldsymbol{x})$. But in practice, we only have access to the finite training set $\mathcal{D}_{\text{train}}$. Hence, we ask the question: "*Can we have infinite training samples drawn from $p_{\text{data}}(\boldsymbol{x})$?*" The answer is likely to be "No", unless we have access to the true data generating process. However, we do have a class of models, known as diffusion models (Sohl-Dickstein et al., 2015; Ho et al., 2020; Song et al., 2021), that can estimate $p_{\text{data}}(\boldsymbol{x})$ very well, and that can generate as many sample as we want. Diffusion models achieve the state of the art performance at data generation, but they lack the

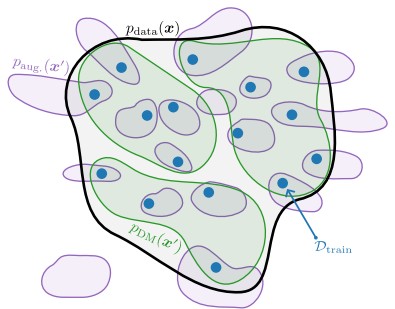

Ideal:
$$\mathcal{L} = \mathbb{E}_{\boldsymbol{x} \sim p_{\text{data}}(\boldsymbol{x})} \left[ \text{ELBO}_{\Theta}(\boldsymbol{x}) \right] \tag{1}$$

Normal Training:
$$\mathcal{L} = \mathbb{E}_{\boldsymbol{x} \sim \mathcal{D}_{\text{train}}} \left[ \text{ELBO}_{\Theta}(\boldsymbol{x}) \right] \tag{2}$$

Augmentation:
$$\mathcal{L} = \mathbb{E}_{\boldsymbol{x} \sim \mathcal{D}_{\text{train}}} \left[ \mathbb{E}_{p_{\text{aug}}(\boldsymbol{x}' \mid \boldsymbol{x})} \left[ \text{ELBO}_{\Theta}(\boldsymbol{x}') \right] \right] \tag{3}$$

DMaaPx (proposed):
$$\mathcal{L} = \mathbb{E}_{\boldsymbol{x}' \sim p_{\text{DM}}(\boldsymbol{x}')} \left[ \text{ELBO}_{\Theta}(\boldsymbol{x}') \right] \tag{4}$$

Figure 1: Left: training distributions for the VAE. Note that $p_{\text{aug}}(\boldsymbol{x}') = \mathbb{E}_{\boldsymbol{x} \sim \mathcal{D}_{\text{train}}}[p_{\text{aug}}(\boldsymbol{x}' \mid \boldsymbol{x})]$ only extrapolates from individual data points $\boldsymbol{x} \sim \mathcal{D}_{\text{train}}$ and has density outside the support of $p_{\text{data}}(\boldsymbol{x})$ (e.g., when flipping the digit "2"). By contrast, the pre-trained diffusion model $p_{\text{DM}}(\boldsymbol{x}')$ can interpolate between data $\boldsymbol{x} \sim \mathcal{D}_{\text{train}}$. Right: corresponding VAE training objectives.

nice properties we want from VAEs, such as semantically meaningful representations. If diffusion models are indeed close estimates for $p_{\text{data}}(\boldsymbol{x})$, we should be able to alleviate the overfitting problem in VAEs using samples from pre-trained diffusion models. In this work, we investigate the effect of modifying the normal training procedure of VAEs by replacing the finite training set $\mathcal{D}_{\text{train}}$ with unlimited samples generated by a pre-trained diffusion model $p_{\text{DM}}(\boldsymbol{x}')$. This idea can be considered as *cross-model-class distillation*, i.e., distilling from a diffusion model to a VAE.

Data augmentation is another method that can generate unlimited data, and it is used to reduce overfitting. However, selecting appropriate augmentations requires human expertise, and the augmented data might inaccurately represent $p_{\text{data}}(\boldsymbol{x})$. Hence, augmentation might lead to training a wrong probabilistic model, even though it can reduce overfitting. Figure 1, discussed further in Section 4, illustrates the relation between the underlying data distribution $p_{\text{data}}(\boldsymbol{x})$, the training set $\mathcal{D}_{\text{train}}$, the augmented training data distribution $p_{\text{aug}}(\boldsymbol{x}')$, and the pre-trained diffusion model $p_{\text{DM}}(\boldsymbol{x}')$.

We empirically show that the new method can indeed alleviate the overfitting issue in VAEs. Specifically, VAEs trained with the new method have better test set performance on estimating the density, on doing approximate inference, and on robustness against adversarial attacks. We also show that we do not need infinite data to gain such generalization performance. As an additional contribution, we publish all the samples we used to train our VAEs, so others do not need to spend compute to train and sample from diffusion models again.

## 2 PERFORMANCE GAPS IN VAES

VAEs model the data distribution $p_{\text{data}}(\boldsymbol{x})$ by assuming that its generative process first draws a latent variable $\boldsymbol{z}$ from $p(\boldsymbol{z})$ and then draws $\boldsymbol{x}$ from $p_{\theta}(\boldsymbol{x} \mid \boldsymbol{z})$ with model parameters $\theta$. Given a training data distribution $p(\boldsymbol{x})$ that approximates $p_{\text{data}}(\boldsymbol{x})$, naive maximum likelihood learning would maximize

$$\max_{\theta} \mathbb{E}_{\boldsymbol{x} \sim p(\boldsymbol{x})} \left[ \log p_{\theta}(\boldsymbol{x}) \right] = \max_{\theta} \mathbb{E}_{\boldsymbol{x} \sim p(\boldsymbol{x})} \left[ \log \int p_{\theta}(\boldsymbol{x} \mid \boldsymbol{z}) \, p(\boldsymbol{z}) \, \mathrm{d}\boldsymbol{z} \right]. \tag{5}$$

Generally, maximizing this likelihood is difficult since we need to integrate over the latent variable $\boldsymbol{z}$. Hence, VAEs turn to an approximate inference method, called variational inference. This method introduces an approximate posterior $q_{\phi}(\boldsymbol{z} \mid \boldsymbol{x})$ within a tractable variational family, and maximizes a lower bound of Eq. (5), known as the evidence lower bound (ELBO; Blei et al. (2017))

$$\log p_{\theta}(\boldsymbol{x}) \geq \mathbb{E}_{\boldsymbol{z} \sim q_{\phi}(\boldsymbol{z} \mid \boldsymbol{x})} \left[ \log p_{\theta}(\boldsymbol{x} \mid \boldsymbol{z}) + \log p(\boldsymbol{z}) - \log q_{\phi}(\boldsymbol{z} \mid \boldsymbol{x}) \right] =: \text{ELBO}_{\Theta}(\boldsymbol{x}), \tag{6}$$

where $\Theta = \{\theta, \phi\}$. In VAEs, the approximate posterior $q_{\phi}(\boldsymbol{z} \mid \boldsymbol{x})$ is usually a Gaussian distribution parameterized by the output of a neural network $f_{\phi}(\boldsymbol{x})$ with weights $\phi$. We call $p_{\theta}(\boldsymbol{x} \mid \boldsymbol{z})$ the *conditional likelihood* to distinguish it from the *likelihood* $p_{\theta}(\boldsymbol{x})$. The distribution of $p_{\theta}(\boldsymbol{x} \mid \boldsymbol{z})$ is also parameterized by the output of a network $g_{\theta}(\boldsymbol{z})$ with weights $\theta$. We often refer to $f_{\phi}(\boldsymbol{x})$ as the inference network (or the encoder) and $g_{\theta}(\boldsymbol{z})$ as the generative network (or the decoder).

Combining Eq. (5) and Eq. (6), we have the training objective of VAEs, i.e., to maximize

$$\mathcal{L} = \mathbb{E}_{\boldsymbol{x} \sim p(\boldsymbol{x})} \left[ \mathrm{ELBO}_\Theta(\boldsymbol{x}) \right]. \tag{7}$$

Ideally, we would like to use $p_{\mathrm{data}}(\boldsymbol{x})$ for $p(\boldsymbol{x})$, but in reality we only have access to $\mathcal{D}_{\mathrm{train}}$. We now discuss three performance metrics for VAEs to evaluate the degree and the impact of overfitting. These metrics are defined in term of gaps, and will be used in the experiment section below.

**Generalization gap.** One signal for overfitting is that a model performs better on the training set $\mathcal{D}_{\mathrm{train}}$ than on the test set $\mathcal{D}_{\mathrm{test}}$, and the test set performance decreases over training epochs. For VAEs, we refer to the difference between training and test set ELBO as the *generalization gap*

$$\mathcal{G}_{\mathrm{g}} = \mathbb{E}_{\boldsymbol{x} \sim \mathcal{D}_{\mathrm{train}}} \left[ \mathrm{ELBO}_\Theta(\boldsymbol{x}) \right] - \mathbb{E}_{\boldsymbol{x} \sim \mathcal{D}_{\mathrm{test}}} \left[ \mathrm{ELBO}_\Theta(\boldsymbol{x}) \right]. \tag{8}$$

Since $\mathcal{D}_{\mathrm{train}}$ and $\mathcal{D}_{\mathrm{test}}$ both consist of samples from the same distribution $p_{\mathrm{data}}(\boldsymbol{x})$, and training maximizes the ELBO on $\mathcal{D}_{\mathrm{train}}$, the ELBO on $\mathcal{D}_{\mathrm{train}}$ is greater than or equal to the ELBO on $\mathcal{D}_{\mathrm{test}}$. Therefore, $\mathcal{G}_{\mathrm{g}} \geq 0$. A smaller $\mathcal{G}_{\mathrm{g}}$ corresponds to a better generalization performance of a VAE.

**Remark** (Test data entropy can also affect the ELBO value). Note that from Eqs. (6) and (7), we have

$$\mathbb{E}_{\boldsymbol{x} \sim p(\boldsymbol{x})} \left[ \mathrm{ELBO}_\Theta(\boldsymbol{x}) \right] \leq \mathbb{E}_{\boldsymbol{x} \sim p(\boldsymbol{x})} \left[ \log p_\theta(\boldsymbol{x}) \right] = -H[p(\boldsymbol{x}), p_\theta(\boldsymbol{x})] \leq -H[p(\boldsymbol{x})], \tag{9}$$

where $H$ denotes the (cross) entropy. Therefore, the ELBO on $\mathcal{D}_{\mathrm{test}}$ can be higher than the ELBO on $\mathcal{D}_{\mathrm{train}}$, if $\mathcal{D}_{\mathrm{train}}$ and $\mathcal{D}_{\mathrm{test}}$ are not drawn from the same distribution, and $\mathcal{D}_{\mathrm{test}}$ has a lower entropy than $\mathcal{D}_{\mathrm{train}}$. Indeed, this phenomenon has been observed in the out-of-distribution setting when testing on a low-entropy data set (Nalisnick et al., 2018). We will refer back to this in Section 5.6.

**Amortization gap.** VAEs use amortized inference, i.e., they set the variational parameters of $q_\phi(\boldsymbol{z} \,|\, \boldsymbol{x})$ to the output of the encoder $f_\phi(\boldsymbol{x})$ for all given $\boldsymbol{x}$. At test time, we can further maximize the ELBO over the individual variational parameters for each $\boldsymbol{x}$, which is more expensive but typically results in a better variational distribution $q^*(\boldsymbol{z} \,|\, \boldsymbol{x})$. We then study the *amortization gap*,

$$\mathcal{G}_{\mathrm{a}} = \mathbb{E}_{\boldsymbol{x} \sim \mathcal{D}_{\mathrm{test}}} \left[ \mathrm{ELBO}_\theta^*(\boldsymbol{x}) \right] - \mathbb{E}_{\boldsymbol{x} \sim \mathcal{D}_{\mathrm{test}}} \left[ \mathrm{ELBO}_\Theta(\boldsymbol{x}) \right] \tag{10}$$

where $\mathrm{ELBO}_\theta^*(\boldsymbol{x}) = \mathbb{E}_{\boldsymbol{z} \sim q^*(\boldsymbol{z} \,|\, \boldsymbol{x})} \left[ \log p_\theta(\boldsymbol{x} \,|\, \boldsymbol{z}) + \log p(\boldsymbol{z}) - \log q^*(\boldsymbol{z} \,|\, \boldsymbol{x}) \right]$. As mentioned before, the encoder $f_\phi(\boldsymbol{x})$ is more susceptible to overfitting than the decoder in VAEs. When the encoder overfits, its inference ability might not generalize to test data, which results in lower ELBO value and larger amortization gap. The amortization gap $\mathcal{G}_{\mathrm{a}}$ is non-negative and a smaller $\mathcal{G}_{\mathrm{a}}$ corresponds to better generalization performance of the inference model (or encoder).

**Robustness gap.** An overfitted encoder $f_\phi(\boldsymbol{x})$ often learns a less smooth function such that a small change in the input space can lead to a huge difference in the output space. Hence, it is easier to construct an adversarial sample $\boldsymbol{x}^{\mathrm{a}} = \boldsymbol{x}^{\mathrm{r}} + \boldsymbol{\epsilon}$ (s.t. $\|\boldsymbol{\epsilon}\| \leq \delta$) from a real data point $\boldsymbol{x}^{\mathrm{r}} \in \mathcal{D}_{\mathrm{test}}$. This is done by maximizing the symmetrized KL-divergence (Kullback & Leibler, 1951) between $q_\phi(\boldsymbol{z} \,|\, \boldsymbol{x}^{\mathrm{r}})$ and $q_\phi(\boldsymbol{z} \,|\, \boldsymbol{x}^{\mathrm{a}})$ within a given attack radius $\delta$ (Kuzina et al., 2022). A successful attack means that the attack reconstruction $\tilde{\boldsymbol{x}}^{\mathrm{a}} = g_\theta(\boldsymbol{z}^{\mathrm{a}})$, $\boldsymbol{z}^{\mathrm{a}} \sim q_\phi(\boldsymbol{z} \,|\, \boldsymbol{x}^{\mathrm{a}})$, is very different from the real data reconstruction $\tilde{\boldsymbol{x}}^{\mathrm{r}} = g_\theta(\boldsymbol{z}^{\mathrm{r}})$, $\boldsymbol{z}^{\mathrm{r}} \sim q_\phi(\boldsymbol{z} \,|\, \boldsymbol{x}^{\mathrm{r}})$, even though the inputs $\boldsymbol{x}^{\mathrm{a}}$ and $\boldsymbol{x}^{\mathrm{r}}$ are similar. Using the image similarity metric MS-SSIM (Wang et al., 2003), we define the *robustness gap* as

$$\mathcal{G}_{\mathrm{r}} = \mathbb{E}_{\boldsymbol{x}^{\mathrm{a}} \sim p(\boldsymbol{x}^{\mathrm{a}} \,|\, \boldsymbol{x}^{\mathrm{r}})} \mathbb{E}_{\boldsymbol{x}^{\mathrm{r}} \sim \mathcal{D}_{\mathrm{test}}} \left[ \mathrm{MS\text{-}SSIM} \left[ \boldsymbol{x}^{\mathrm{r}}, \boldsymbol{x}^{\mathrm{a}} \right] - \mathrm{MS\text{-}SSIM} \left[ \tilde{\boldsymbol{x}}^{\mathrm{r}}, \tilde{\boldsymbol{x}}^{\mathrm{a}} \right] \right]. \tag{11}$$

Note that a higher MS-SSIM corresponds to a more similar data pair. Hence, MS-SSIM $[\boldsymbol{x}^{\mathrm{r}}, \boldsymbol{x}^{\mathrm{a}}]$ is greater than or equal to MS-SSIM$[\tilde{\boldsymbol{x}}^{\mathrm{r}}, \tilde{\boldsymbol{x}}^{\mathrm{a}}]$, and the gap $\mathcal{G}_{\mathrm{r}}$ is a non-negative value. A more robust VAE has a higher MS-SSIM$[\tilde{\boldsymbol{x}}^{\mathrm{r}}, \tilde{\boldsymbol{x}}^{\mathrm{a}}]$ than the less robust one. Therefore, a smaller $\mathcal{G}_{\mathrm{r}}$ corresponds to a more robust VAE. For more details on the attack see Appendix A.

## 3 RELATED WORK

We group related work into using diffusion models as data sources and attempts to closing the three performance gaps. Work related to data augmentation and distillation is discussed in Section 4.

**Use samples from pre-trained diffusion models.** There are many recent attempts to solve various tasks with data generated by diffusion models. Azizi et al. (2023) fine-tuned a text-to-image diffusion model on ImageNet, generated state-of-the-art samples with class labels, and trained a classifier on the samples. Their result shows that the classifier trained on generated data does not outperform the classifier trained on real data. In the adversarial training setting, using generated data by diffusion models shows significant improvements on classification robustness (Croce et al., 2021; Wang et al., 2023). Tian et al. (2023) found that the visual representations learned from samples generated by text-to-image diffusion models outperform the representations learned by SimCLR and CLIP. Alemohammad et al. (2023) trained new diffusion models with samples from previously trained diffusion models, and they found that their sample quality and diversity progressively decrease. In this work, we find that using diffusion models as data sources improves the performance of VAEs.

**Improve generalization, amortized inference, and robustness in VAEs.** Cremer et al. (2018) study the amortization gap in VAEs, and they notice that overfitting in the encoder is one of the contributing factors of the gap, and it hurts the generalization of VAEs. Many subsequent works try to close the amortization gap by introducing new inference techniques or procedures (Marino et al., 2018; Shu et al., 2018; Zhao et al., 2019). To close the generalization gap and reduce encoder overfitting, Zhang et al. (2022) propose to freeze the decoder after a certain amount of training steps, but further train the encoder by using reconstruction samples as part of the training data. As for adversarial robustness in VAEs, Kuzina et al. (2022) propose to defend a pre-trained VAE by running MCMC during inference to move $z$ towards "safer" regions in the latent space. Our proposed method can be used on top of these existing methods, since it does not require changing the original inference procedure. It also takes into account all three gaps at the same time.

## 4   DIFFUSION MODEL AS A $p_{\text{data}}(\boldsymbol{x})$

In this section, we introduce a new method for reducing overfitting in VAEs (Section 4.1). We also discuss how the new method is fundamentally different from naive data augmentation (Section 4.2), and how it can be understood from a cross-model-class distillation perspective (Section 4.3).

### 4.1   PROPOSED METHOD

The ideal training objective for VAEs is to maximize $\mathbb{E}_{\boldsymbol{x}\sim p_{\text{data}}(\boldsymbol{x})}\left[\text{ELBO}_{\Theta}(\boldsymbol{x})\right]$ (see Eq. (1) in Figure 1). However, in practice, we only have $\mathcal{D}_{\text{train}}$ as a finite approximation of $p_{\text{data}}(\boldsymbol{x})$. Hence, we normally maximize $\mathbb{E}_{\boldsymbol{x}\sim\mathcal{D}_{\text{train}}}\left[\text{ELBO}_{\Theta}(\boldsymbol{x})\right]$ (see Eq. (2)) to train a VAE, which can lead to overfitting. Rather than focusing on model architectures or training techniques as in prior works, we aim to mitigate overfitting by seeking a better approximation for $p_{\text{data}}(\boldsymbol{x})$ than $\mathcal{D}_{\text{train}}$. Here, we make two assumptions: first, the training data distribution should fulfill two criteria; it should be

(1) **a continuous distribution**, i.e., we can sample unlimited data to avoid overfitting; and

(2) **an accurate approximation of** $p_{\text{data}}(\boldsymbol{x})$, i.e., we are indeed modeling $p_{\text{data}}(\boldsymbol{x})$ rather than some different distribution (in practice, it needs to be an accurate model of $\mathcal{D}_{\text{train}}$).

Our second assumption is that a good diffusion model[1] that has been pre-trained on $\mathcal{D}_{\text{train}}$ satisfies these two criteria: (1) we can generate unlimited samples from it, and (2) its training objective is designed to model $p_{\text{data}}(\boldsymbol{x})$, allowing us to generate samples with state-of-the-art quality across various data types. Therefore, we investigate training VAEs using a pre-trained diffusion model $p_{\text{DM}}(\boldsymbol{x}')$ instead of $\mathcal{D}_{\text{train}}$ as an approximation of $p_{\text{data}}(\boldsymbol{x})$, i.e., to maximize $\mathbb{E}_{\boldsymbol{x}\sim p_{\text{DM}}(\boldsymbol{x}')}\left[\text{ELBO}_{\Theta}(\boldsymbol{x}')\right]$ (see Eq. (4)). We denote this method DMaaPx, short for "Diffusion Model as a $p_{\text{data}}(\boldsymbol{x})$".

Figure 1 illustrates the intuition behind this idea. The blue dots represent the finite data set $\mathcal{D}_{\text{train}}$. They are i.i.d. samples from the underlying data distribution $p_{\text{data}}(\boldsymbol{x})$ (shown by the dark-edged region). The green regions represent the distribution learned by $p_{\text{DM}}(\boldsymbol{x}')$. We use areas, not dots, to highlight that $p_{\text{DM}}(\boldsymbol{x}')$ models a continuous distribution that can generate infinitely many samples.

Note that diffusion models for data types other than images are less explored and might not accurately approximate $p_{\text{data}}(\boldsymbol{x})$. Hence, diffusion models might not satisfy criterion (2). Moreover,

---

[1]More precisely, an *unconditional* diffusion model, as opposed to a conditional one such as Stable Diffusion.

due to the data processing inequality, information on $p_{\text{data}}(\boldsymbol{x})$ captured by a diffusion model that was trained on $\mathcal{D}_{\text{train}}$ cannot exceed the information contained in $\mathcal{D}_{\text{train}}$. In reality, state-of-the-art diffusion models are not able to fit $\mathcal{D}_{\text{train}}$ perfectly. Indeed, many recent works observe that in both image and text settings, training generative models from generated data leads to worse performance overall (Alemohammad et al., 2023; Shumailov et al., 2023). Hence, the continuity we gain by replacing $\mathcal{D}_{\text{train}}$ with $p_{\text{DM}}(\boldsymbol{x}')$ is not for free, we lose a small amount of information about $\mathcal{D}_{\text{train}}$.

## 4.2 DIFFERENCE BETWEEN DATA AUGMENTATION AND DMAAPX

Data augmentation[2] pursues a similar goal as the proposed DMaaPx as both approaches aim to increase the quantity and diversity of training data. The primary distinction between them is in their accuracy in approximating $p_{\text{data}}(\boldsymbol{x})$, as shown in Table 1. This can be attributed primarily to two key factors. Firstly, typical data augmentation techniques generate new training points by conditioning on a

Table 1: Training distributions for VAEs (see Figure 1), and whether they are (1) continuous and (2) an accurate approximation of $p_{\text{data}}(\boldsymbol{x})$.

| approx. by | $\mathcal{D}_{\text{train}}$ | $p_{\text{aug}}(\boldsymbol{x}')$ | $p_{\text{DM}}(\boldsymbol{x}')$ |
|---|---|---|---|
| (1) continuous | ✗ | ✓ | ✓ |
| (2) accurate | ✓ | ✗ | ✓ |

*single* original data point. Thus, $p_{\text{aug}}(\boldsymbol{x}') = \mathbb{E}_{\boldsymbol{x} \sim \mathcal{D}_{\text{train}}}[p_{\text{aug}}(\boldsymbol{x}' \,|\, \boldsymbol{x})]$ where $p_{\text{aug}}(\boldsymbol{x}' \,|\, \boldsymbol{x})$ generates a training point $\boldsymbol{x}'$ by applying one or more random transformations (e.g., padding, cropping, flipping (He et al., 2016), translation or even learned rotation and cutout (Cubuk et al., 2019)) to a single original data point $\boldsymbol{x}$. By contrast, in the proposed DMaaPx, each training data point $\boldsymbol{x}' \sim p_{\text{DM}}(\boldsymbol{x}')$ is drawn from a diffusion model that was trained on the entire dataset $\mathcal{D}_{\text{train}}$. As a consequence, each training data point in DMaaPx is effectively conditioned on the full training set $\mathcal{D}_{\text{train}}$.

Secondly, the random transformations used for $p_{\text{aug}}(\boldsymbol{x}' \,|\, \boldsymbol{x})$ in traditional data augmentation are drawn from a manually curated catalog. This catalog is heavily based on prior assumptions regarding *invariances* in the data type under consideration, which can introduce bias. In practice, one has to make assumptions and decide whether the (unknown) true data distribution $p_{\text{data}}(\boldsymbol{x})$ is invariant under the considered transformations. For instance, with images, we assume invariance to minor translations, hue shifts, and zooms. This may result in problems of (i) not modeling the *full* extend of the distribution or (ii) modeling density *outside* the true data distribution. Figure 1 depicts both: problem (i) corresponds to "empty" space between areas of $p_{\text{aug}}(\boldsymbol{x}')$; problem (ii) corresponds to density of $p_{\text{aug}}(\boldsymbol{x}')$ outside of $p_{\text{data}}(\boldsymbol{x})$. The proposed DMaaPx eliminates these explicit assumptions, which makes the method more resilient against human bias (but less interpretable).

In summary, while traditional data augmentation techniques introduce diversity based on *invariances* about the data generative process, the proposed DMaaPx uses an expressive generative model to extrapolate from the *empirical* diversity of the data.

## 4.3 A CROSS-MODEL-CLASS DISTILLATION PERSPECTIVE

The proposed DMaaPx can also be viewed from a distillation perspective (Hinton et al., 2015). Distillation describes the process of transferring knowledge from a large model to a small one. In practice, distillation is often used because a smaller model is less expensive to be deployed in production. Here we consider a more subtle usage of distillation, i.e., transferring knowledge between models designed with different modeling assumptions or structures. We refer to this as *cross-model-class distillation*, and the conventional usage of distillation as *within-model-class distillation*. There are models that have been designed with useful structures which cannot be fully exploited if trained naively on $\mathcal{D}_{\text{train}}$. Cross-model-class distillation creates auxiliary training data that helps us train such models to achieve the desired performance. For instance, in the diffusion model literature, numerous studies attempt to distill the multi-step diffusion process into a single-step generative model (Salimans & Ho, 2021; Luhman & Luhman, 2021; Liu et al., 2023; Song et al., 2023). While both types of distillation seek to transfer knowledge from a source to a target model, cross-model-class distillation emphasizes more on enhancing functionalities that are unique to the target model rather than mirroring the capabilities shared with the source model.

---

[2]Using samples from generative models for training is sometimes considered as a kind of data augmentation in the context of supervised learning (Yang et al., 2022). In the present work, we deliberately separate generative models from the broader sense of data augmentation, and we consider data augmentation in a narrow sense such that the augmented data is an output of some transformation conditioned on an single $\boldsymbol{x} \in \mathcal{D}_{\text{train}}$.

Our proposed DMaaPx belongs to cross-model-class distillation, i.e., it distills diffusion models to VAEs. The goal of DMaaPx is not to rival diffusion models in sample quality, but rather to improve the desirable functionalities of VAEs such as representation learning. From this viewpoint, DMaaPx fundamentally differs from approaches that train VAEs on samples produced by VAEs, or diffusion models on outputs of diffusion models (Alemohammad et al., 2023; Shumailov et al., 2023). Such approaches can be categorized as within-model-class distillation.

## 5 EXPERIMENTS

In this section, we introduce the experimental setup and evaluate the three performance gaps (see Section 2) of the proposed method. The exact gap values are provided in Appendix B. We further investigate whether we need infinite training data in the proposed method.

### 5.1 EXPERIMENTAL SETUP

**Training data.** We evaluate our method on three popular datasets: MNIST (LeCun et al., 1998), FashionMNIST (Xiao et al., 2017), and CIFAR-10 (Krizhevsky et al., 2009). As a preparation, we train a diffusion model $p_{\mathrm{DM}}(\boldsymbol{x}')$ which will be used to generate training data for VAEs on each training set $\mathcal{D}_{\mathrm{train}}$. We use the implementation of diffusion models by Karras et al. (2022). Further details and samples from the three pre-trained diffusion models can be found in Appendix C.

**VAE architectures.** We assume fixed standard Gaussian priors $p(\boldsymbol{z}) = \mathcal{N}(\boldsymbol{0}, \mathbf{I})$ for all datasets. For the conditional likelihood $p_\theta(\boldsymbol{x} \,|\, \boldsymbol{z})$, we use a Bernoulli distribution for binarized MNIST, a diagonal Gaussian distribution with a fixed variance for grayscale FashionMNIST, and a discretized mixture of logistics (MoL; Salimans et al. (2017)) for CIFAR-10. For the inference model $q_\phi(\boldsymbol{z} \,|\, \boldsymbol{x})$, we use diagonal Gaussian distributions with means and variances output from the inference network for all datasets. For more details on network architectures and hyperparameters see Appendix D.

**Baselines.** We compare VAEs trained with our proposed DMaaPx against three baseline models trained on: (i) repetitions of $\mathcal{D}_{\mathrm{train}}$ ("Normal Training"); (ii) carefully tuned augmentation for $\mathcal{D}_{\mathrm{train}}$ ("Aug.Tuned"); and (iii) plausible augmentation for images in general ("Aug.Naive"). Note that "Aug.Naive" is not tuned to a given $\mathcal{D}_{\mathrm{train}}$ and can result in out-of-distribution data, e.g. a horizontally flipped digit "2" for MNIST. This mimics situations that arise in augmenting other data modalities, where the choice of transformation is not as obvious as for images. More details on the applied augmentation can be found in Appendix E. Generally, when documenting the training progress, the term "epoch" typically refers to one complete pass of $\mathcal{D}_{\mathrm{train}}$. For DMaaPx, this term is not applicable since it can sample unlimited data from $p_{\mathrm{DM}}(\boldsymbol{x}')$. Therefore, we measure training progress of DMaaPx in "effective epochs". An "effective epoch" represents one pass through sampled training data of size $|\mathcal{D}_{\mathrm{train}}|$. We train all models for 1000 (effective) epochs.

### 5.2 GENERALIZATION GAP

Figure 2 shows both ELBOs evaluated on $\mathcal{D}_{\mathrm{train}}$ (dashed) and $\mathcal{D}_{\mathrm{test}}$ (solid) for all three datasets. The difference between these two lines is the generalization gap $\mathcal{G}_{\mathrm{g}}$ (Eq. (8)). We observe that our proposed DMaaPx (green) has the highest ELBO on $\mathcal{D}_{\mathrm{test}}$, and the smallest generalization gap compared to both normal training and data augmentation. VAEs trained on the augmented data show less improvements on test ELBO and generalization gap than DMaaPx. This implies that VAEs trained with DMaaPx approximate the underlying distribution $p_{\mathrm{data}}(\boldsymbol{x})$ better than those trained on $\mathcal{D}_{\mathrm{train}}$ solely, or on augmented data. The small generalization gap of DMaaPx means that training ELBOs can be used as accurate predictions for final performance. Given that data from pre-trained diffusion models and augmentation is not an inherently more accurate representation of $p_{\mathrm{data}}(\boldsymbol{x})$ than $\mathcal{D}_{\mathrm{train}}$, improvements in the test ELBOs suggest that overfitting in VAEs is more detrimental than using a somewhat distorted, but larger and more diverse, dataset.

### 5.3 AMORTIZATION GAP

The amortization gap, defined in Eq. (10), evaluates the encoder's inference performance by comparing test ELBOs of the amortized variational parameters of $q_\phi(\boldsymbol{z} \,|\, \boldsymbol{x})$ to those from individually

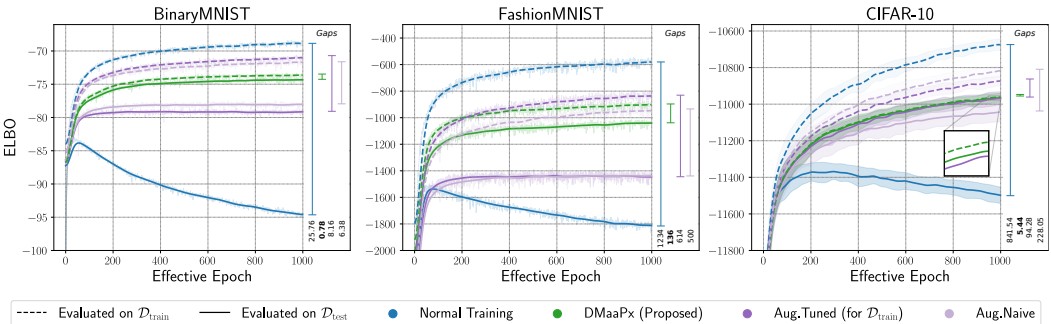

Figure 2: Generalization performance: ELBOs of models trained with Eqs. (2)-(4), evaluated on $\mathcal{D}_{\text{train}}$ and $\mathcal{D}_{\text{test}}$. DMaaPx (proposed) consistently has the best test performance and smallest gap.

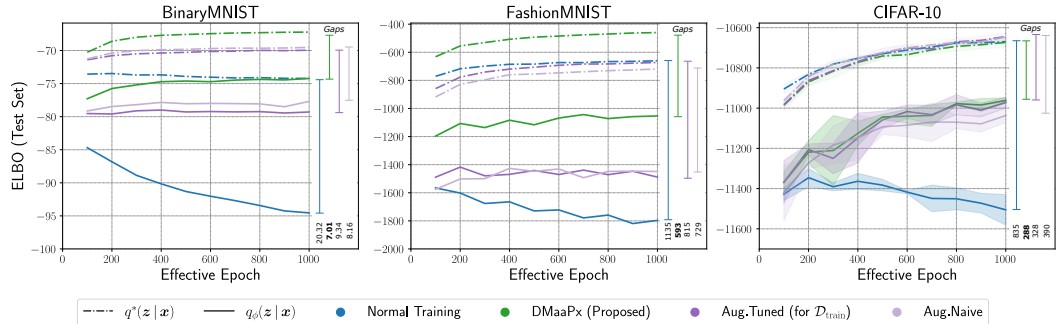

Figure 3: Amortization gap: ELBOs with $q_\phi(z \mid x)$ from the inference network (solid) and with iteratively optimized $q^*(z \mid x)$ (dashdot), evaluated on $\mathcal{D}_{\text{test}}$. Left and center: DMaaPx significantly reduces the amortization gap (Eq. (10)) compared to normal training and data augmentation. Among the dashdot lines, DMaaPx has the highest performance, indicating that it also helps learning a better decoder. Right: DMaaPx and augmentation are tied and outperform normal training.

optimized variational parameters of $q^*(z \mid x)$. Figure 3 shows test set ELBOs from $q_\phi(z \mid x)$ and $q^*(z \mid x)$ for the three datasets with values reported every 100 effective epochs.

The figure illustrates that the ELBOs for normal training using $q_\phi(z \mid x)$ (solid blue) decline with more training epochs, while those using $q^*(z \mid x)$ (dashdot blue) remain stable or even increase. A decline in test set performance across epochs signals overfitting. By using $q^*(z \mid x)$ and excluding the encoder, test performance stabilizes across epochs, indicating that the primary source of overfitting in VAEs is the encoder. This aligns with the findings of Cremer et al. (2018).

Figure 3 shows that DMaaPx outperforms normal training and data augmentation in both, the size of the amortization gap and the ELBO value for BinaryMNIST and FashionMNIST. Additionally, the increase of ELBOs with $q^*(z \mid x)$ (dashdot) suggests that DMaaPx also improves the decoder. On CIFAR-10, DMaaPx and augmentation similarly outperform normal training.

## 5.4 ROBUSTNESS OF REPRESENTATIONS (ROBUSTNESS GAP)

The robustness gap, defined in Eq. (11), looks at similarities between real and the corresponding adversarial samples (MS-SSIM$[x^r, x^a]$) and between their respective reconstructions (MS-SSIM$[\tilde{x}^r, \tilde{x}^a]$). A successful attack achieves low MS-SSIM$[\tilde{x}^r, \tilde{x}^a]$ despite high MS-SSIM$[x^r, x^a]$. See Appendix A for details on the attack construction in our experiments.

In Figure 4, we see that DMaaPx consistently matches or surpasses normal training across all three datasets. It also exceeds augmentation on BinaryMNIST and CIFAR-10. Meanwhile, VAEs trained with augmentation display inconsistent results: they outperform both DMaaPx and normal training on FashionMNIST, but fall behind on BinaryMNIST and CIFAR-10, demonstrating that augmentation is more difficult to tune than DMaaPx (training the diffusion model requires less manual effort).

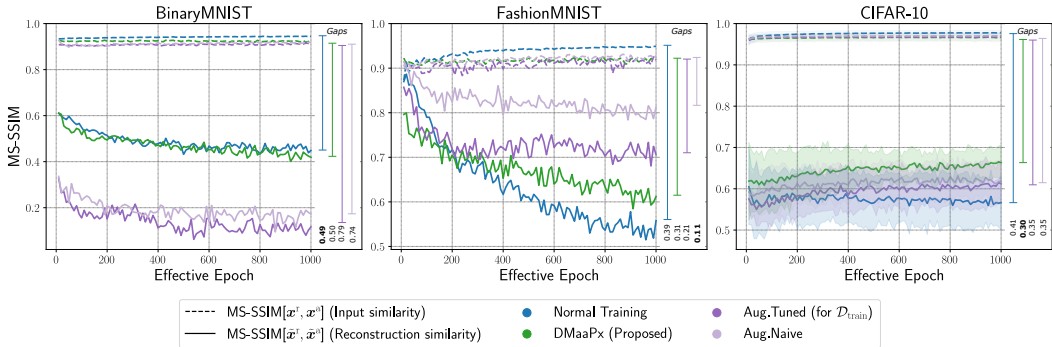

Figure 4: Adversarial robustness: similarities of reconstructions (solid) for similar but adversarially chosen inputs (see dashed line). DMaaPx is consistently either on par or better than normal training whereas augmentation is significantly worse than normal training for BinaryMNIST and CIFAR-10.

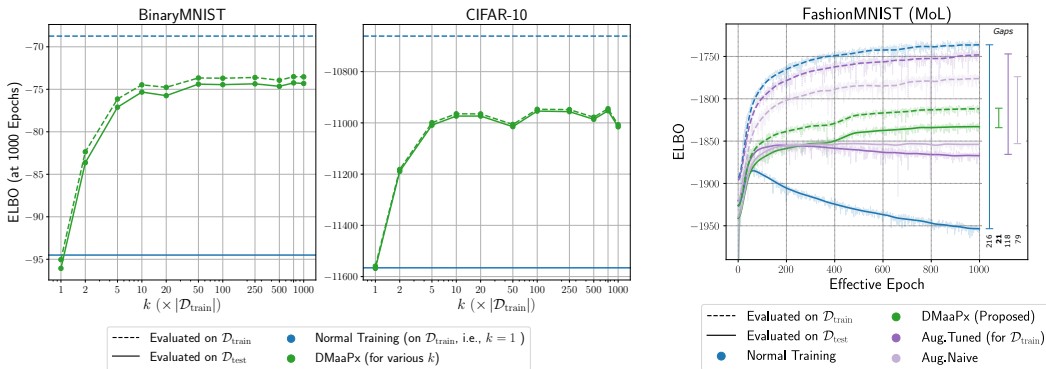

Figure 5: Generalization performance as a function of the amount $k$ of training data sampled from the diffusion model. Horizontal blue lines show baseline performance (VAE trained directly on $\mathcal{D}_{\text{train}}$). All VAEs were trained for 1000 effective epochs. $k \approx 10$ seems to suffice.

Figure 6: Generalization performance for FashionMINST with MoL likelihood. We observe similar behavior to center panel in Figure 2, which uses a Gaussian likelihood.

## 5.5 IS THE "UNLIMITED DATA PLAN" A RIPOFF?

With the pre-trained diffusion model in DMaaPx, we can train VAEs with unlimited samples from $p_{\text{DM}}(\boldsymbol{x}')$, enhancing performance as demonstrated above. While generating a large amount of samples from diffusion models is feasible, it still requires substantial computation. Therefore, we further explore: "*Do we really need infinite number of samples?*" The answer, reassuringly, is "*No*".

Figure 5 shows the generalization performance of DMaaPx on BinaryMNIST and CIFAR-10 where the training data for VAEs is restricted to $k \times |\mathcal{D}_{\text{train}}|$, with $k$ ranging from 1 to 1000. After $k$ effective training epochs, samples start repeating. All models are trained on 1000 effective epochs. Horizontal blue lines represent the generalization gap of normal VAE training (on $\mathcal{D}_{\text{train}}$ and $k = 1$) at epoch 1000 from Figure 2. For $k = 1$, DMaaPx slightly underperforms on BinaryMNIST but matches normal training on CIFAR-10. The ELBO plateaus for $k \geq 10$, indicating samples roughly 10 times the size of $\mathcal{D}_{\text{train}}$ offer similar generalization to samples 1000 times larger.

## 5.6 ABLATION AND FURTHER DETAILS

In this section we present ablations on different conditional likelihoods, compare the two augmentation strategies considered, and discuss the difference between training ELBO and ELBO on $\mathcal{D}_{\text{train}}$.

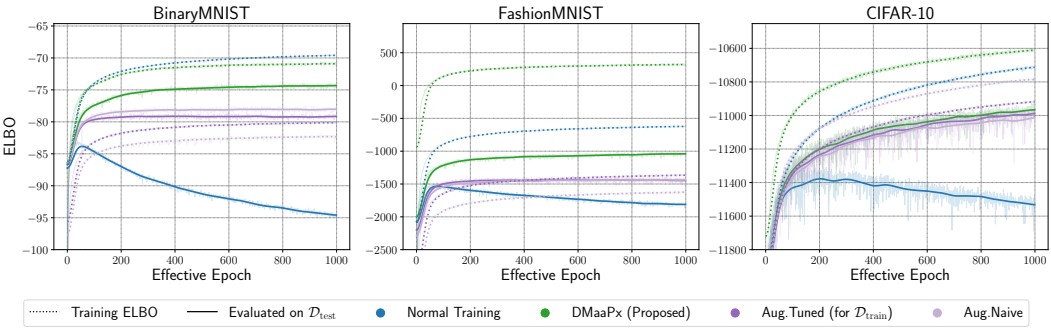

Figure 7: ELBO evaluted on the distribution that is actually used for training (dotted, see Eqs. (2)-(4)). For augmentations, the test ELBO (solid) is higher than the training ELBO (dotted) in the left two panels, which is an artifact of different entropies of the distributions, see Remark.

**Different conditional likelihoods.** VAEs' modeling assumptions for the conditional likelihood $p_\theta(x \mid z)$ often differ based on data or use case. While a Gaussian likelihood is used for applications that focus on low reconstruction error (e.g., lossy data compression), an MoL likelihood is used if the density of the data matters (e.g., generative modeling or lossless data compression). Our experiments in Sections 5.2-5.4 cover three likelihoods: Bernoulli for BinaryMNIST, Gaussian for FashionMNIST, and MoL for CIFAR-10. Figure 6 also evaluates MoL for FashionMNIST, and we observe similar behaviour as in its Gaussian counterpart in Figure 2 (center). In summary, DMaaPx is less prone to overfitting than normal training and augmentation, for all investigated likelihoods.

**Tuned and naive augmentation.** To fairly assess DMaaPx against augmentation, we design two sets of augmentation: Aug.Tuned (tailored to each $\mathcal{D}_{\text{train}}$) and Aug.Naive (general for images). They perform similarly overall in Figures 2-4. However, Aug.Naive outperforms Aug.Tuned in generalization on BinaryMNIST and FashionMNIST, and in robustness across all datasets. This is surprising as naive augmentation might produce out-of-distribution data, like a horizontally flipped digit "2", potentially impairing performance. Thus, designing augmentation can be labor-intensive.

**Training ELBO vs. ELBO on $\mathcal{D}_{\text{train}}$.** Figure 7 shows the ELBOs analogous to Figure 2, but the dotted lines plot the ELBO on the actual training distribution (e.g., on samples from $p_{\text{DM}}(x')$ for DMaaPx). The point of this plot is to warn that comparisons between ELBOs under such different distributions are not meaningful, and should not be used to calculate the generalization gap. For example, note that the plot would suggest a negative generalization gap for data augmentation (purple) on BinaryMNIST. This is consistent with the remark on page 3: since the ELBO is bounded by the negative entropy of the distributions on which it is evaluated, evaluating it on two different distributions with different entropies exhibits differences unrelated to the generalization gap.

## 6 CONCLUSION

We investigate how overfitting of VAEs can be addressed by training them on samples from a diffusion model that was pre-trained on the training dataset. Our assumption is that, unlike in supervised learning, VAE training requires training data that accurately matches the data generative process. We therefore contrasted our approach to traditional data augmentation methods, which might not accurately model the data generative process. Our results show significant reduction in generalization gaps, improved test ELBOs, and enhanced adversarial robustness. Future work should challenge the above assumption and investigate whether one can further improve VAE performance by designing a generative model that specifically for cross-model-class distillation.

In a broader sense, our work explores ways of increasing the quantity and diversity of training data in situations where one cares about the underlying data distribution. Future work should also expand this research beyond VAEs, in particular as prior work found that recursive distillation within a diffusion model hurts performance (Alemohammad et al., 2023). Additional work should explore applying our method to other data types, such as structured data like molecules or time series.

**Reproducibility Statement.** All code necessary to reproduce the results in this paper is provided in the supplementary materials. We will also publish the samples generated by our pre-trained diffusion models for the DMaaPx experiments after the reviewing process.

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

## A  DETAILS ON ADVERSARIAL ATTACK

We follow Kuzina et al. (2022) and construct an unsupervised encoder attack that optimizes the pertubation $\epsilon$ to incur the largest possible change in $q_\phi(\cdot \mid \boldsymbol{x})$,

$$\epsilon = \underset{\|\epsilon\|_\infty \leq \delta}{\arg\max} \ \mathrm{SKL}\left[q_\phi(\cdot \mid \boldsymbol{x}^\mathrm{r} + \epsilon) \,\|\, q_\phi(\cdot \mid \boldsymbol{x}^\mathrm{r})\right] \tag{12}$$

where SKL denotes the symmetric Kullback-Leibler divergence (Kullback & Leibler, 1951). We optimize $\epsilon$ for $n^\epsilon$ iterations with projected gradient descent utilizing a learning rate of $\eta$. The robustness gap (see Section 2) is computed over $n^\mathrm{r}$ real images and $n^\mathrm{a}$ random seeds. The exact hyperparameters can be found in Table 2.

Table 2: Hyperparameters for unsupervised encoder attack.

|              | BinaryMNIST | FashionMNIST | CIFAR-10 |
|--------------|-------------|--------------|----------|
| $n^\mathrm{r}$ | 50          | 50           | 20       |
| $n^\mathrm{a}$ | 10          | 10           | 10       |
| $n^\epsilon$ | 50          | 50           | 100      |
| $\eta$       | 1.0         | 1.0          | 1.0      |
| $\delta$     | 0.1         | 0.1          | 0.05     |

## B  QUANTITATIVE RESULTS ON PERFORMANCE GAPS

Table 3 assigns quantitative values to the visual evidence in Figure 2 (generalization gap), Figure 3 (amortization gap), and Figure 4 (adversarial robustness gap).

Table 3: Quantitative values of the performance gaps visualized in the main text (generalization gap: Figure 2; amortization gap: Figure 3; robustness gap: Figure 4). Bold numbers indicate the smallest gap within a dataset.

|                   |                 | generalization gap ($\mathcal{G}_\mathrm{g}$, Eq. (8)) | amorization gap ($\mathcal{G}_\mathrm{a}$, Eq. (10)) | robustness gap ($\mathcal{G}_\mathrm{r}$, Eq. (11)) |
|-------------------|-----------------|--------------------|-----------------|----------------|
| Binary MNIST      | Normal Training | 25.76              | 20.32           | **0.49**       |
|                   | DMaaPx (ours)   | **0.78**           | **7.01**        | 0.50           |
|                   | Aug.Tuned       | 8.16               | 9.34            | 0.79           |
|                   | Aug.Naive       | 6.38               | 8.16            | 0.74           |
| Fashion MNIST     | Normal Training | 1234.50            | 1135.89         | 0.39           |
|                   | DMaaPx (ours)   | **136.57**         | **593.39**      | 0.31           |
|                   | Aug.Tuned       | 614.93             | 815.52          | 0.21           |
|                   | Aug.Naive       | 500.33             | 729.83          | **0.11**       |
| CIFAR-10          | Normal Training | 841.54             | 835.86          | 0.41           |
|                   | DMaaPx (ours)   | **5.44**           | **288.82**      | **0.30**       |
|                   | Aug.Tuned       | 94.28              | 328.08          | 0.35           |
|                   | Aug.Naive       | 228.05             | 390.25          | 0.35           |

## C  DIFFUSION MODEL

We follow the setup of Karras et al. (2022) for the design and training of our diffusion model. However, we do not use the proposed augmentation pipeline during training.

We train the diffusion model on $200,000,000$ images that are sampled randomly (with replacement) from the training dataset. Each model is trained on 8 NVIDIA A100 40GB GPUs for approximately 2.5 days.

We utilized the deterministic second-order sampler as proposed by Karras et al. (2022) with 18 integration steps. Each sampled image utilizes a unique initial seed. We sample on a single NVIDIA A100 40GB GPU. Sampling $50,000$ images takes approximately 25 to 30 minutes.

Figure 8 shows samples from the diffusion models trained. On CIFAR-10 we report a FID score of 3.9537. Scores on BinaryMNIST and FashionMNIST are ommited as those are not widely reported and heavily depend on preprocessing (Song et al., 2021).

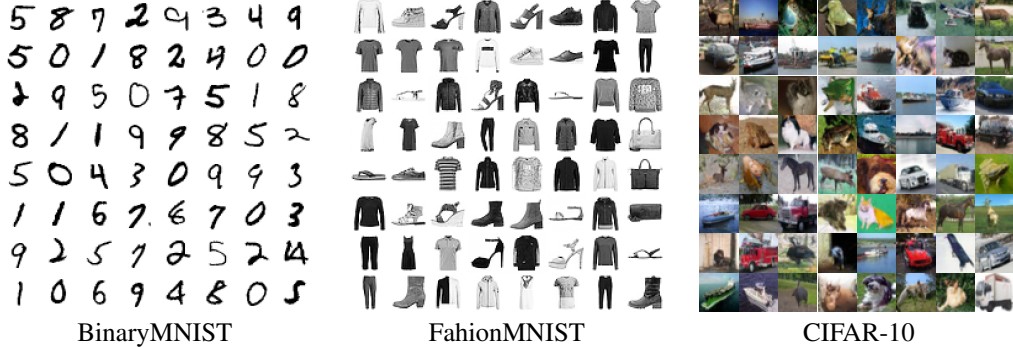

BinaryMNIST            FahionMNIST            CIFAR-10

Figure 8: Samples of the diffusion models trained on BinaryMNIST (LeCun et al., 1998), Fashion-MNIST (Xiao et al., 2017), and CIFAR-10 (Krizhevsky et al., 2009).

## D DETAILS ON VAE ARCHITECTURES

This section provides a detailed description of the VAE models utilized throughout the paper. We consider a fully-connected architecture and a residual architecture (He et al., 2016). Table 4 gives more details on the likelihood model and architecture. For BinaryMNIST and FashionMNIST, we chose the hyperparameters of the VAE models by consulting the literature. For CIFAR-10, we manually tried out a few hyperparameters, and chose an architecture where overfitting occurs, as we are investigating how to alleviate overfitting in VAEs only from the training data.

The fully-connected architecture maps from an input dimension of $32^2$ to a hidden dimension of $512$. After a hidden layer mapping from $512$ to $512$, the output is mapped to a latent variable of dimension $16$. The decoder mirrors the encoder and maps the latent variable of dimension $16$ via three layers to the original input size.

The residual architecture maps the input by two convolutional layers (kernel size: 4, stride: 2, padding: 1), two residual layers, and another convolutional layer to a latent dimension of $64$. The residual connection is made up of two convolutional layers where the first one applies a kernel of size 3 (kernel size: 3, stride: 1, padding: 1) and the second one applies a kernel of size 1 (kernel size: 1, stride: 1, padding: 0) All convolutional layers do not use any biases and are followed by BatchNorm (Ioffe & Szegedy, 2015). The decoder mirrors the architecture of the encoder.

Table 4: Details on VAE architectures ordered by dataset. MoL refers to the discretized mixture of logistics likelihood model (Salimans et al., 2017).

| dataset | likelihood | architecture |
|---|---|---|
| BinaryMNIST | Bernoulli | fully-connected |
| FashionMNIST | fixed-variance Gaussian | fully-connected |
| FashionMNIST | MoL | fully-connected |
| CIFAR-10 | MoL | residual network |

# E  AUGMENTATION

We use the augmentation pipeline originally proposed for GAN training following Karras et al. (2020). Each specific augmentation is applied with a probability of $b \in \{0.1, 0.12\}$. For each dataset we compare two sets of specific augmentations.

1. The hyperparameters for each specific augmentation are tuned by hand with the goal of imitating the data generating distribution that produced the dataset.

2. We use a naive set of specific augmentations that is targeted to image datasets.

Table 5 lists naive augmentation for BinaryMNIST, FashionMNIST, and CIFAR-10. Table 6 lists augmentation tuned to the BinaryMNIST and the FashionMNIST dataset. Table 7 lists augmentation tuned to the CIFAR-10 datset.

Table 5: List of specific augmentations applied to BinaryMNIST, FashionMNIST and CIFAR-10. We refer to this set as "naive" augmentation as it is targeted towards images in general (and not towards specific datasets). Each specific augmentation is applied with probability $b$.

| augmentation | description and hyperparameters |
|---|---|
| horizontal flip | flip an image horizontally |
| translation | translate an image in $x$ and $y$ direction for $t \in \{0, 1, 2, 3\}$ pixels |
| scaling | scale an image by $2^{\sigma_{\text{scale}}}$ with $\sigma_{\text{scale}} \in [0, 0.2]$ |
| rotation | rotate an image by $d$ degrees with $d \in [0, 10]$ |
| anisotropic scaling | do anisotropic scaling with scale $2^{\sigma_{\text{aniso-scale}}}$ ($\sigma_{\text{aniso-scale}} \in [0, 0.2]$) |
| anisotropic rotation | do anisotropic rotation with a probability of $0.5$ |
| brightness | change the brightness of an image by $\sigma_{\text{brightness}} \in [0, 0.2]$ |
| contrast | change the contrast of an image by $2^{\sigma_{\text{contrast}}}$ where $\sigma_{\text{contrast}} \in [0, 0.25]$ |
| hue | change the hue by rotation of $r_{\text{hue}}$ with $r_{\text{hue}} \in [0, 0.25 \cdot \pi]$ |
| saturation | change the saturation of an image by $2^{\sigma_{\text{saturation}}}$ where $\sigma_{\text{saturation}} \in [0, 0.5]$ |

Table 6: List of specific augmentations applied to BinaryMNIST and FashionMNIST. The set is tuned towards BinaryMNIST and FashionMNIST. Each specific augmentation is applied with probability $b$.

| augmentation | description and hyperparameters |
|---|---|
| translation | translate an image in $x$ and $y$ direction for $t \in \{0, 1, 2, 3\}$ pixels |
| scaling | scale an image by $2^{\sigma_{\text{scale}}}$ with $\sigma_{\text{scale}} \in [0, 0.15]$ |
| rotation | rotate an image by $d$ degrees with $d \in [0, 10]$ |
| anisotropic scaling | do anisotropic scaling with scale $2^{\sigma_{\text{aniso-scale}}}$ ($\sigma_{\text{aniso-scale}} \in [0, 0.15]$) |
| anisotropic rotation | do anisotropic rotation with a probability of $0.4$ |

Table 7: List of specific augmentations applied to CIFAR-10. The set is tuned towards CIFAR-10. Each specific augmentation is applied with probability $b$.

| augmentation | description and hyperparameters |
|---|---|
| horizontal flip | flip an image horizontally (applied with probability 1) |
| vertical flip | flip an image vertically |
| scaling | scale an image by $2^{\sigma_{\text{scale}}}$ with $\sigma_{\text{scale}} \in [0, 0.2]$ |
| rotation | rotate an image by $d$ degrees with $d \in [0, 360]$ |
| anisotropic scaling | do anisotropic scaling with scale $2^{\sigma_{\text{aniso-scale}}}$ ($\sigma_{\text{aniso-scale}} \in [0, 0.2]$) |
| anisotropic rotation | do anisotropic rotation with a probability of $0.5$ |

# F  PRACTICAL EVALUATION OF VAEs ON THREE TASKS

The improvements of generalization performance, amortized inference and robustness of VAEs have direct impacts on their applications. In this section, we evaluate three popular tasks of VAEs based on whether a task involves only the encoder, the decoder, or both as in (Xiao & Bamler, 2023): (a) representation learning (i.e., using only the encoder); (b) data reconstruction (i.e., using both the encoder and the decoder); and (c) sample generation (i.e., using only the decoder).

**Representation learning (with classification as the downstream task).** We evaluate the representation learning performance by classification accuracies on the mean $\boldsymbol{\mu}$ of $q_\phi(\boldsymbol{z} \mid \boldsymbol{x})$ for each $\boldsymbol{x}$. First, we find the learned representations $\boldsymbol{\mu}$ for all data points in the CIFAR-10 test set. Afterwards, we split them into two separate subsets. We use one subset to train the classifier, and test it on the other subset. Our experiments include four different classifiers: logistic regression, a support vector machine (Boser et al., 1992) with radial basis function kernel (SVM-RBF), a SVM with linear kernel (SVM-L), and $k$-nearest neighbors (kNN) with $k = 5$. Table 8 (representation learning; **RL**) shows the resulting test accuracies across all models considered. We find that VAEs trained with DMaaPx (in bold) outperform other models on average, which highlights that the task of representation learning benefits from the smaller gaps evaluated in Section 5.

**Data reconstruction.** Tasks such as lossy data compression (Ballé et al., 2017) rely on the reconstruction performance of VAEs. We evaluate the reconstruction performance of VAEs trained on CIFAR-10 using the peak signal-to-noise ratio (PSNR; higher is better). Table 8 (reconstruction; **RC**) shows that DMaaPx outperforms others on average.

**Sample generation.** We evaluate the quality of samples generated by VAEs trained on CIFAR-10 with the methods explained in the main text (Normal Training, DMaaPx, Aug.Naive, Aug.Tuned). We report Fréchet Inception Distance (Heusel et al., 2017) (FID; lower is better) and Inception Score (Salimans et al., 2018) (IS; higher is better). Table 8 (sample quality; **SQ**) shows that DMaaPx slightly outperforms the others when sample quality is measured in FID, but Normal Training performs better when sample quality is measured in IS.

Overall, VAEs trained with DMaaPx show improvements for representation learning and data reconstruction, and perform similarly to normal training on sample quality. At the same time, VAEs trained with both augmentations seem to have slightly worse performance for representation learning and sample generation, and perform similarly on the reconstruction task when compared to normal training. The results of DMaaPx in the table is consistent with our claim that the proposed method mainly fixes the encoder, which affects representation learning and reconstruction but not sample quality. Additionally, Theis et al. (2016) show that a generative model with good log-likelihood (i.e., high test ELBO in the case of a VAE) does not necessarily produce great samples.

Table 8: Evaluation of downstream applications of VAEs on CIFAR-10: representation learning with classification as the downstream task (**RL**), reconstruction (**RC**), and sample quality (**SQ**). Results are averaged over 3 random seeds. Note that most differences are smaller than the standard deviations. See Appendix F for a discussion of the results.

|    |    | Normal Training | DMaaPx (ours) | Aug.Naive | Aug.Tuned |
|----|----|---|---|---|---|
|    | log. reg. ($\uparrow$) | $0.370 \pm 0.018$ | $\mathbf{0.383 \pm 0.018}$ | $0.359 \pm 0.004$ | $0.361 \pm 0.014$ |
| **RL** | SVM-RBF ($\uparrow$) | $0.427 \pm 0.014$ | $\mathbf{0.438 \pm 0.015}$ | $0.421 \pm 0.004$ | $0.420 \pm 0.016$ |
|    | SVM-L ($\uparrow$) | $0.367 \pm 0.015$ | $\mathbf{0.380 \pm 0.014}$ | $0.365 \pm 0.005$ | $0.366 \pm 0.022$ |
|    | kNN ($\uparrow$) | $0.325 \pm 0.006$ | $\mathbf{0.327 \pm 0.035}$ | $0.300 \pm 0.004$ | $0.299 \pm 0.028$ |
| **RC** | PSNR ($\uparrow$) | $16.087 \pm 0.042$ | $\mathbf{16.370 \pm 0.195}$ | $16.105 \pm 0.017$ | $15.924 \pm 0.205$ |
| **SQ** | FID ($\downarrow$) | $219.256 \pm 16.124$ | $\mathbf{219.081 \pm 14.894}$ | $237.238 \pm 43.218$ | $240.898 \pm 11.072$ |
|    | IS ($\uparrow$) | $\mathbf{1.818 \pm 0.155}$ | $1.614 \pm 0.076$ | $1.656 \pm 0.047$ | $1.612 \pm 0.083$ |

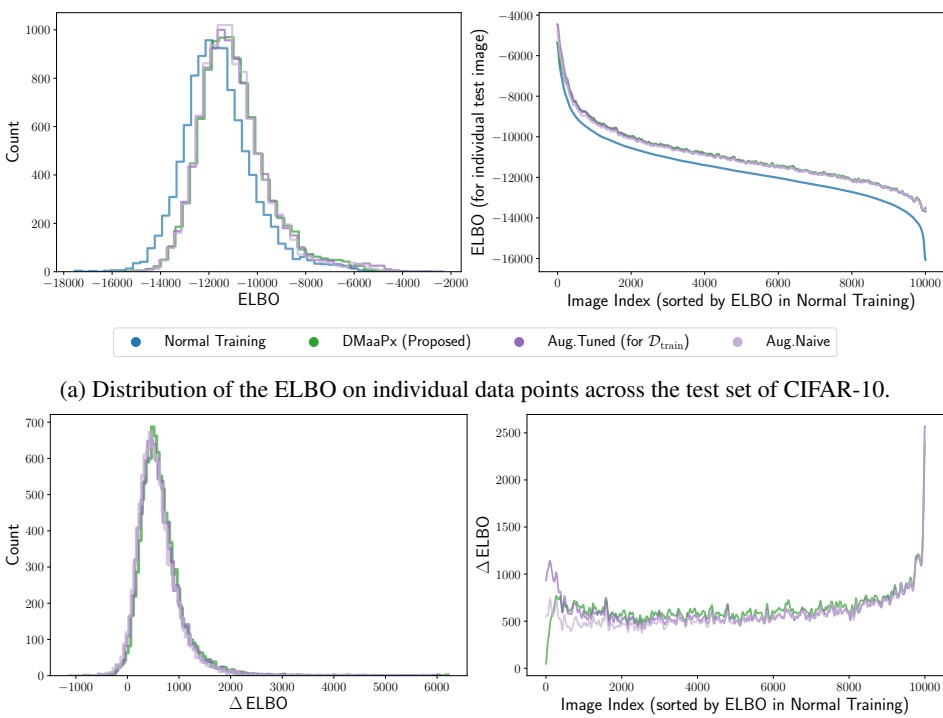

(a) Distribution of the ELBO on individual data points across the test set of CIFAR-10.

(b) Distribution of the ELBO differences when compared to Normal Training, i.e., $\Delta\mathrm{ELBO} := \mathrm{ELBO}_p[\boldsymbol{x}] - \mathrm{ELBO}_{\mathcal{D}_{\mathrm{train}}}[\boldsymbol{x}]$, where $p$ is $p_{\mathrm{DM}}(\boldsymbol{x}')$ if evaluated on the VAE trained with DMaaPx or the corresponding $p_{\mathrm{aug}}(\boldsymbol{x}')$ if evaluated on the VAE trained with Aug.Tuned and Aug.Naive.

Figure 9: Individual ELBO evaluated on CIFAR-10 test set. Left: histograms for ELBO and ELBO differences ($\Delta\mathrm{ELBO}$) on individual image. Right: ELBO and $\Delta\mathrm{ELBO}$ values for individual image. Data are ordered by ELBO values of Normal Training from high (index 1) to low (index 10000).

## G   ELBOS ON INDIVIDUAL TEST IMAGES

In this section, we investigate the distribution of the ELBO values on individual data points of the CIFAR-10 test set (that has a size of $10,000$), as one might be curious whether DMaaPx or augmentations only improve VAEs on a subset of the training data.

Figure 9 (a, left) shows a histogram of ELBO values for all methods. We find that the distribution of ELBO values shifts to the right (i.e., ELBOs are larger) when comparing Normal Training to other methods. We do not see any significant differences between DMaaPx, Aug.Tuned, and Aug.Naive. Figure 9 (a, right) shows the ELBO evaluated on individual test images. The test images are ordered and indexed based on their ELBO values with Normal Training from high (index 1) to low (index 10000). Both DMaaPx, Aug.Tuned, and Aug.Naive perform similarly better compared to the model with Normal Training across all test images. We can verify the same finding when plotting the differences between DMaaPx, Aug.Tuned, Aug.Naive and Normal Training. Figure 9 (b) shows the distribution of differences in a histogram (left) and for individual test images (right). We find that our method improves the ELBO on almost all (99.9 %) of the test points (see Figure 9 (b, right)).

Overall, the ELBO improvement by both DMaaPx and augmentations is observed across all test images, and we could not identify a subset of test data points where the improvement is particularly small or large when compared to Normal Training.

