# OpenReview forum: "Upgrading VAE Training With Unlimited Data Plans Provided by Diffusion Models"
_ICLR.cc/2024/Conference — Submitted to ICLR 2024_

### Official Review · Reviewer_FgWw · 2023-10-28

**Soundness:** 3 good
**Presentation:** 3 good
**Contribution:** 2 fair
**Rating:** 5
**Confidence:** 3

**Summary:**

This work is motivated by the fact that VAEs learn meaningful representations however tend to overfit to the training samples. Motivated by various shortcomings of augmentation, they propose to use diffusion models as a surrogate method to "complete" or improve the quality of the data, given their success at approximating probability distributions. The paper then conducts experiments studying the generalization gap.

**Strengths:**

- The paper takes a rather simple idea which is to bootstrap data using Diffusion models and shows how it can be used to effectively improve VAEs.
- The paper is very easy to follow, with background and reference material laid out quite well.
- The generalization, amortization and robustness quantities are well defined and nicely motivated.

**Weaknesses:**

- The concern in the paper is the lack of any theoretical investigation into why these methods work. While this is difficult to do in general, it would better if there was any mention of it at least with simplified models or any assumptions.
- Diffusion methods can also be interpreted as a form of augmentation. For example you can use them to generate more examples in different resolutions or even "image-to-image" translation methods. Thus, I think the argument of treating augmentations visually different in Figure 1 is not very clear. While it mentions that augmentations are not "accurate", I think this will largely depend on the domain/data. I also think another aspect to consider is the computational time.`
- I believe this paper is highly dependent on the accuracy of the diffusion model and while a theoretical investigation would further clarify this, I think it is worth mentioning. For example, in tabular data settings, diffusion models may not work so effectively or perhaps when data to train the diffusion model itself is scarce. Noting that diffusion models are quite data hungry, I'm not sure if it will be a great idea to use them to bootstrap in general.

**Questions:**

“amortized inference, and adversarial robustness” - can you explain what these are and how VAEs are used. It may be obvious for amortised inference however unclear for adversarial robustness. I think it is a good idea to introduce those problems before and how VAEs are used for them.

I think the statement: “Therefore, Gg ≥ 0” is quite strong however this is not guaranteed. I understand this is what we expect however it could be the case that the test data is very simple to learn and the training data is complicated, resulting in a smaller ELBO over train. I would amend this statement to say “we anticipate after training G_g \geq 0$ as opposed to keeping it as a statement by itself.

Have you tried other types of generative models to bootstrap a VAE? For example, what if you use a GAN since they can be quite robust. Also, GANs can be used to generate synthetic augmentation and thus lie in the "accurate" region in between augmentation and diffusion.

Other:
Full stop page 1: "robustness For generative modeling" -> "robustness. For generative modeling"
“but they lack of the” -> “but they lack the”

---

> ### Author Response · Authors · 2023-11-17
> **Reply to Reviewer FgWw (R4) from the authors**
>
> Thank you for your detailed feedback! We are glad that you think the “paper is easy to follow” and the three gaps/metrics are “well defined and nicely motivated”. We address your questions below, and we updated the PDF accordingly.
>
> > **Q1** - [...] lack of any theoretical investigation [...]
>
> Indeed a general theory would be difficult. In Section 4.2, we mention that data augmentation generates new data points by sampling $\mathbf x'\sim p_\text{aug}(\mathbf x'|\mathbf x)$ conditioned on a _single_ data point $\mathbf x$, whereas our method generates new data points conditioned on the _entire_ training set.
>
> For a simplified model, consider a data set where the underlying data distribution $p_\text{data}(\mathbf x)$ is a mixture of Guassians. Traditional augmentation would be limited to adding small noise to each data point (ensuring that it stays within its mode) when $p_\text{data}(\mathbf x)$ is unknown. But if we have knowledge about $p_\text{data}(\mathbf x)$, e.g. if the mixture components lie on a regular grid, then we can use discrete translations as augmentations. By contrast, a diffusion model trained on the entire data set could detect such properties automatically.
>
> > **Q2** - [diffusion models can be interpreted as data augmentation, e.g., in image-to-image translation]
>
> Good point! _Conditional_ diffusion models are indeed similar to traditional data augmentation. However, we propose to use an _unconditional_ diffusion model, i.e., a model that samples random data without taking an input. We clarified this as a footnote in Section 4.1 in the paper.
>
> > **Q3** - [...] diffusion models are quite data hungry
>
> Thank you for bringing this up as it highlights why our results are nontrivial! We show that, whether or not a given model is data hungry is a relative statement. While the training sets were _too small_ for fitting good VAEs, as all baseline VAEs in our experiments strongly overfit, the same training sets are _large enough_ to fit useful diffusion models. One might obtain better diffusion models with larger training sets, but our diffusion models were evidently already at least good enough to be useful within our proposed method.
>
> The fact that this observation is counterintuitive highlights why it should be communicated to a broader audience: one might think diffusion models require lots of data, and this is true if the goal is to generate high-quality samples. But our paper demonstrates a surprising new application for diffusion models in the regime where one does have too little data to train a good VAE.
>
> > **Q4** - [computational time]
>
> Note that our method only increases training time but not test time, which is more important in many applications. For further comments, please refer to item (2) in our “Overall response” above.
>
> > **Q5** - [...] [results are] highly dependent on the accuracy of the diffusion model [...]
>
> Indeed, as we discuss in the last paragraph of Section 4.1, diffusion models on other data types than images are less explored, and we might not have accurate models. But research in this area currently receives a lot of attention, and we expect that our method is compatible with any models proposed in the future.
>
> > **Q6** - [explain amortized inference and adversarial robustness in the context of VAEs]
>
> Amortized inference is a fundamental principle of VAEs. We review this term briefly at the beginning of the paragraph “Amortization Gap” (Section 2). Expanding on this explanation, amortized inference refers to the fact that VAEs use a neural network (the “inference network”) to map data points to latent representations which speeds up training and deployment. One can remove the inference network and instead obtain latent representations by iteratively optimizing the ELBO for each data point (as we do for $q^*$ in Section 5.3), but this is very expensive.
> The end of Section 2 discusses in detail the kind of adversarial robustness that we analyze. We are happy to provide further explanation if something is not clear.
>
> > **Q7** - [$\mathcal G_\text{g} \geq 0$ is not guaranteed]
>
> The statement $\mathcal G_\text{g} \geq 0$ relies on the assumption that training and test data come from the same distribution (see below Eq. 8). The remark below Eq. 8 discusses the case where this assumption is violated, and where the test data is “simpler” than the training data (i.e., it comes from a distribution with lower entropy). As discussed below Eq. 9, this can indeed lead to an inversion of the ELBOs (i.e., to $\mathcal G_\text{g} < 0$).
>
> > **Q8** - [...] other types of generative models to bootstrap a VAE?
>
> A major reason why diffusion models have become so popular (and why we use them here) is because they avoid difficulties in the training objectives of GANs which makes them easier to train.
>
> Thank you again for your review! We would appreciate it if you could let us know whether our above responses resolve your concerns, and change your assessment of our work accordingly.

---

### Official Review · Reviewer_nb6g · 2023-10-30

**Soundness:** 3 good
**Presentation:** 3 good
**Contribution:** 3 good
**Rating:** 5
**Confidence:** 4

**Summary:**

The focus of the paper is around how to prevent overfitting of the encoder part of the VAE through using a diffusion model to sample from the data distribution. The authors analyze the performance based on various metrics and conclude that having access to a diffusion model for samples helps to prevent the overfitting problem.

**Strengths:**

Overall the paper is a novel way in combining a cutting edge generative model in conjugation with a VAE. I think the most interesting result is that it is counter to some of the previous work in the area that suggested sampling data could not produce good enough training data. The method could help to prevent overfitting for when VAEs are used for tasks where a latent representation is needed. Overall the paper is well written and I liked the Figure 1 which helped to simplify the story of the paper.

**Weaknesses:**

The concept presented in the paper is intriguing. However, I struggled to discern scenarios where the proposed method would be essential in the context of a VAE. It's conceivable that there are applications in biological data or reinforcement learning where one might need to map an image to a latent representation and subsequently utilize this representation. Yet, the paper primarily focuses on image generation, a task for which the diffusion model already excels. The computational overhead introduced by the proposed method seems substantial. What advantages does it offer to justify this overhead? I would appreciate further evidence demonstrating situations where a diffusion model sampler enhances the VAE's performance for specific, necessary tasks.

**Questions:**

-	Figure 2: I found this a bit confusing we want the difference to the smallest but the way you plot both its hard to see which method is doing the best given the scale, perhaps you could replot just showing difference. Why is there a zoom in on the CIFAR-10 plot what is that supposed to show?
-	Figure 3: Similarly here can we plot the gap it seems the augmented approach perhaps does just as well but hard to see.
-	Am I right that all plots are just one training curve how did you pick hyperparameters for each method? Did you try running for different seeds to get some randomness in the results?
-	Do you have samples from your VAE plotted or any sort of test metric on how they do for your method versus the baselines?

---

> ### Author Response · Authors · 2023-11-17
> **Reply to Reviewer nb6g (R3) from the authors**
>
> Thank you very much for your thorough review. We are pleased that you think our proposed method is “novel”, “the concept presented in the paper is intriguing”, and the results are “interesting”.
> We address your questions below, and we updated the PDF accordingly.
>
>
> > **Q1** - [...] paper primarily focuses on image generation [...] for which the diffusion model already excels [...]
>
> We agree that diffusion models already excel at image generation and that there is little benefit to using VAEs for this task. However, while our method uses a generative model (diffusion model) to generate images, our evaluations do not focus on image generation but instead on density estimation and representation learning, which are domains where VAEs provide an advantage over diffusion models. Also, as discussed in the introduction, the goal of our method is mainly to improve the encoder of the VAE, which does not play a role in sample generation. For more clarifications how our evaluations are not about generative performance, please refer to item (1) in our “Overall response regarding sample quality and computational cost” posted above.
>
> We agree, however, that our paper focuses on the image domain since this is the domain where both diffusion models and VAEs are best understood. While diffusion models on other data types are currently less explored, research in this area currently receives a lot of attention, and we expect that our method is compatible with any models proposed in the future.
>
> > **Q2** -  [...] further evidence demonstrating [...] VAE's performance for specific, necessary tasks.
>
> We agree that Sections 5.2-5.4 evaluate the performance of the VAE somewhat indirectly.
> Therefore, we added more direct evaluations of practical downstream applications in Appendix F (_new_), specifically representation learning tasks (classifying latent representations), reconstruction tasks, and (for completeness) also data generation tasks.
> While performance differences are indeed small, our proposed method enhances the VAE’s performance for all tasks and all metrics except when measuring sample quality (SQ) by inception score (IS).
> This is consistent with our claim that the proposed method mainly fixes the encoder, which affects representation learning and reconstruction but not sample quality.
>
> > **Q3** -  The computational overhead [...] seems substantial.
>
> We provide numbers for the computational cost in Appendix C.
> We stress that our method only increases training time but not test time, which is more important in many applications.
> Also, the generated samples can be reused when training VAEs with different configurations for cross-validation.
> For further comments, please refer to item (2) in our “Overall response regarding sample quality and computational cost”.
>
> > **Q4** -  Figure 2: [...] it’s hard to see which method is doing the best [...] Figure 3: [...] it seems like the augmented approach does just as well but hard to see
>
> Thank you for the suggestion. We added annotations to Figures 2, 3, 4, and 6 that highlight the gaps and show their numerical values. We hope that these additional annotations make it easy to see that our proposed method achieves the smallest gaps everywhere except for the robustness gap in BinaryMNIST and FashionMNIST (see Table 3 in Appendix B). The reason why we plot ELBO values rather than gaps in Figures 2, 3, 4, and 6 is because both the ELBO values and the gaps are important in practice, and plotting only the gaps would conceal the absolute values.
>
> > **Q5** - [Figure 2:] Why is there a zoom in on the CIFAR-10 plot what is that supposed to show?
>
> We only show this inset because the gap here is so small that it might otherwise be difficult to see that the plot does indeed show two separate green lines.
>
> > **Q6** -  [...] how did you pick hyperparameters for each method?
>
> Thank you for raising this important point. Hyperparameters are listed in Appendix D. For BinaryMNIST and FashionMNIST, we chose the hyperparameters of the VAE models by consulting the literature. For CIFAR-10, we manually tried out a few hyperparameters, and chose an architecture where overfitting occurs, as we are investigating how to alleviate overfitting in VAEs only from the training data. We have updated the Appendix D with this information.
>
> > **Q7** -  [...] running for different seeds [...]
>
> Thank you very much for this important point! You are right that the plots were covering one seed. We re-ran CIFAR-10 experiments with three different random seeds and updated the Figures 2, 3, and 4 with means and standard deviation, which show consistent results across different random seeds. Experiments for the other data set are still running, and we will update the paper as soon as they finish.
>
> Thank you again for your review! We would appreciate it if you could let us know whether our above responses resolve your concerns, and change your assessment of our work accordingly.

---

> > ### Comment · Reviewer_nb6g · 2023-11-20
> > **Response to rebuttal**
> >
> > I thank the authors for addressing my concerns, I appreciate the new experiments added and the additional of multiple seeds for the core experiments.

---

### Official Review · Reviewer_nWgS · 2023-11-01

**Soundness:** 3 good
**Presentation:** 3 good
**Contribution:** 3 good
**Rating:** 5
**Confidence:** 4

**Summary:**

The author shows that utilizing samples from the diffusion model can mitigate issues on VAE and improve its generalization, amortized inference, and robustness.

**Strengths:**

They use diffusers as data augmentation tools to generate informative data for VAE training. These images help reduce the generalization gap,  amortization gap, and robustness gap.

**Weaknesses:**

1. According to figure5, it needs at least 2 times training data to be better than the normal training and 10 times data to reach the best performance. How are the computation costs comparing DMaaPx with aug  tuned and aug naive, to let the model achieve the same/similar performance? (in terms of test ELBO).
2. In your table 3, the method effectively reduced all three gaps the most for Binary MNIST and Fashion MNIST but not CIFAR10. Can you explain why it fails on CIFAR10? It looks to me that when the dataset becomes complex, your method loses its advantage. To convince the audience, you should include more datasets.

**Questions:**

See the weakness above and this additional question below:
1. In figure 2, CIFAR10, the test data’s curves seem to keep increasing. For both augtuned and DMaaPx. If trained longer, what would be the trend?

---

> ### Author Response · Authors · 2023-11-17
> **Reply to Reviewer nWgS (R2) from the authors**
>
> Thank you very much for your review! We address your questions below, and we updated the PDF accordingly.
>
> > **Q1** - According to figure5, it needs at least 2 times training data to be better than the normal training and 10 times data to reach the best performance. How are the computation costs comparing DMaaPx with aug tuned and aug naive, to let the model achieve the same/similar performance? (in terms of test ELBO).
>
> We provide computational costs in Appendix C. In brief, for any reasonable amount of samples from the diffusion models, the computational overhead of DMaaPx mainly comes from *training* the diffusion model and not from sampling from it. Therefore, we see little benefit in sampling less than about $10 \times |\mathcal D_\text{train}|$ data points, at which point sampling more has diminishing returns.
>
> On the wider point of computational cost, note that our method only increases training time but not test time, which is more important in many applications. Also, samples can be reused when training VAEs with different configurations for cross-validation. For further comments, please refer to item (2) in our overall response above.
>
> > **Q2** - In your table 3, the method effectively reduced all three gaps the most for Binary MNIST and Fashion MNIST but not CIFAR10. Can you explain why it fails on CIFAR10? It looks to me that when the dataset becomes complex, your method loses its advantage. To convince the audience, you should include more datasets.
>
> Thank you very much for this comment! Unfortunately Table 3 was showing wrong values (the table was transposed). We apologize for this inconvenience and we uploaded a new version of the paper that fixes this issue. The correct version of the table shows that our method keeps its advantage and does not fail on CIFAR-10. To make it easier to see the gaps, we also added their numerical values to the right margins of Figures 2, 3, 4, and 6 in the main paper.
>
> > **Q3** - In figure 2, CIFAR10, the test data’s curves seem to keep increasing. For both augtuned and DMaaPx. If trained longer, what would be the trend?
>
> Thank you for this question.
> We agree that the ELBOs for data augmentation and our method are not fully converged for CIFAR-10.
> The main message of this plot is that, for normal training, one should have stopped after about 200 epochs since the model already starts to overfit at this point. By contrast, we do not see any overfitting with both DMaaPx and traditional data augmentation for 5x more training epochs.
>
> We are training the models for 2000 epochs on CIFAR-10 at the moment, and we will include the results and a plot as soon as they finish. We already observe in the existing plot that the generalization gap for data augmentation starts to widen when our proposed method still has a negligible gap. We therefore expect that the gap in our method either stays constant (as it does for BinaryMNIST and FashionMNIST) or is the last one to widen.
>
> Thank you again for your review! We hope that our response and the fixed Table 3 resolve any remaining doubts about our paper, and would appreciate it if you update the assessment accordingly.

---

> > ### Comment · Reviewer_nWgS · 2023-11-18
> >
> > Thanks. I really appreciate your reply and additional experiments. Your answer solved my previous questions. However, after taking a closer look at your appendix, I have concern about the influence of paper. In the evaluation of donwstream applications of VAE (RL, RC, SQ), Your method does not have large enough difference in terms of all scores, even compared with the normal training. So for now I'll keep my score.

---

> > > ### Author Response · Authors · 2023-11-20
> > > **Clarification from the authors**
> > >
> > > Thank you very much for being so responsive and for keeping an open mind. We agree that improvements on downstream tasks, while consistent across many metrics, are small in magnitude. We added these additional evaluations as a response to other reviewers’ feedback. We realize that the title of our paper may set wrong expectations, and we are open to rephrasing it to “*Investigating* VAE training…” (as allowed by the [updated author guide](https://iclr.cc/Conferences/2024/AuthorGuide)).
> > >
> > > Indeed, we see the main contribution of our paper not in proposing a better training method but more in *investigating* new ways of addressing the known overfitting problem of VAEs, with a focus on changes to the training data rather than changes to the model architecture or loss function. We do see that the generalization gap shrinks significantly (Figure 2) even though we used an off-the-shelf diffusion model. We clarified in Section 4.1 and in the conclusion that we make the assumptions that (i) cross-model-class distillation requires a generative model that satisfies the two criteria in Table 1, and that (ii) diffusion models satisfy these two criteria. But we hope that our results inspire research that challenges both assumptions, e.g., to develop new generative models that are specifically designed for cross-model-class distillation for VAEs, as this field is much less explored than distillation within discriminative models.

---

### Official Review · Reviewer_ZW6a · 2023-11-02

**Soundness:** 2 fair
**Presentation:** 3 good
**Contribution:** 1 poor
**Rating:** 3
**Confidence:** 4

**Summary:**

In this work, authors propose to distill knowledge from diffusion models to VAEs. Authors show that this approach has some effect on the generalisation of VAE and it’s robustness against adversarial attacks

**Strengths:**

- The proposed idea seems to be novel
- The submission is well written and easy to follow
- The literature review is deep and well presented

**Weaknesses:**

- It is known that diffusion models do not necessarily represent the whole training distribution, also in the image domain (as mentioned in the submission). Could it be that VAE better aligned to the subset of training data that was generated by diffusion? Is the distribution of ELBO on the test-set similar for both approaches?
- In the submission it is claimed that the main goal is to “improve the desirable functionalities of VAEs such as representation learning”, while the evaluation only tackles the quality of data modeling task. It’d be interesting to compare the samples quality between diffusion model from which the knowledge was distilled to the VAE.
- The proposed method has tremendous computational cost that is not evaluated. Training VAE for 1000 effective epochs requires sampling 1000 * $|D_{train}|$ samples from diffusion model, or as presented in Figure 5 at least 10 *$|D_{train}|$ samples.  So the training is at least a few orders of magnitude slower because of sampling, not to mention the cost of training the diffusion model itself.
- The results suggest that distilling knowledge from diffusion models to VAEs introduce some performance gain for simple datasets like MNIST and FMnist, while for slightly more complex dataset - CIFAR10 the performance is comparable to the standard training with augmentations.

**Questions:**

- In section 4.2 there is a discussion on difference between training VAEs on augmented data and training them on data generated from diffusion model. The authors state that training with augmentations may result in problems. However, at the same time diffusion models are usually also trained with the same data augmentation techniques. Does it mean that in this case it was omitted?
- Are the results presented in Figures 2 and 6 statistically significant? The differences in some cases are extremely small, while it seems like results from a single run (judging from the background plot)

---

> ### Author Response · Authors · 2023-11-17
> **Reply to Reviewer ZW6a (R1) from the authors**
>
> Thank you for your insightful feedback! We are glad that you think the proposed idea is “novel”, and the paper is “well written and easy to follow”! We address your questions below, and we updated the PDF accordingly.
>
> > **Q1** - [The diffusion model might focus on a specific subset of the training data.] Is the distribution of the ELBO on the test-set similar for both approaches?
>
> Thank you for this interesting question. We updated the paper and now provide an analysis of the distribution of the ELBO in Appendix G (_new_). We find that our method improves the ELBO on almost all (99.9 %) of the test points (Figure 9 (b)), and we couldn’t identify a subset of test data points where the improvement was particularly small or large.
>
>
> > **Q2** - [...] it is claimed that the main goal is to “improve the desirable functionalities of VAEs such as representation learning”, while the evaluation only tackles the quality of data modeling task.
>
> We believe that there is a misunderstanding, as our experiments in Sections 5.3 and 5.4 do evaluate representation learning. For VAEs, “representation learning” refers to the (output of) the encoder. Section 5.3 evaluates the amortization gap, where a small amortization gap corresponds to a good encoder. Section 5.4 evaluates adversarial robustness. While there are various forms of robustness in VAEs, our experiments specifically evaluate robustness of the encoder, i.e., we test whether an unnoticeable change of the input image can trick the encoder into outputting a semantically different representation. We recognize that this can be easy to miss since we evaluate the success of an attack back in image space.
>
> However, based on reviewer feedback, we added more direct evaluations of practical downstream applications of VAEs in Appendix F (_new_), specifically representation learning tasks (classifying latent representations), reconstruction tasks, and (for completeness) also data generation tasks. While performance differences are indeed small, our proposed method achieves best performance for all tasks and all metrics except when measuring sample quality (SQ) by inception score (IS). This is consistent with our claim that the proposed method mainly fixes the encoder, which affects representation learning and reconstruction but not sample quality.
>
> > **Q3** - [compare sample quality between diffusion model and VAE]
>
> We kindly refer the reviewer to item (1) in our “Overall response regarding sample quality and computational cost” above, which addresses the question of sample quality. In brief, we see little value in using VEAs for purely data generative downstream tasks as diffusion models are known to produce much better samples. However, VAEs can be used for many other downstream tasks that require semantically meaningful representations, which diffusion models don’t provide. Indeed, our paper focuses on the encoder of the VAE since this is the one that tends to overfit.
>
> > **Q4** - [computational cost]
>
> We provide numbers for the computational cost in Appendix C. Note that our method only increases training time but not test time, which is more important in many applications. Also, samples can be reused when training VAEs with different configurations for cross-validation. For further comments, please refer to item (2) in our overall response above.
>
> > **Q5** - [performance gains only on MNIST and FMNIST, but hardly for CIFAR10]
>
> Please note that, as we stress in Section 5.2, we are interested not only in the test ELBOS but also in the generalization _gap_ (Eq. 8): small generalization gaps are very useful as they imply that ELBOs evaluated on the training set can be used for predicting the final performance of the VAE. The proposed DMaaPx has a significantly smaller generalization gap than other methods in all settings.
>
>
> > **Q6** - diffusion models are usually also trained with the same data augmentation techniques. Does it mean that in this case it was omitted?
>
> Thank you for bringing this up. You are right! We indeed omitted data augmentation when training the diffusion model (see details in Appendix C) to not let any augmentation leak to our generated samples.
>
> > **Q7** - Are the results presented in Figures 2 and 6 statistically significant? [...] it seems like results from a single run
>
> Thank you for this valuable point! In the updated version of the paper, we now show means and standard deviations in Figures 2, 3, and 4 on CIFAR-10. We find that our results are consistent across different random seeds. Experiments for the other data sets ~~and for the amortization gap~~ are still running, and we will update the paper as soon as they finish.
>
> Thank you again for your review. We would appreciate it if you could let us know whether our above responses resolve your concerns, and change your assessment of our work accordingly.

---

> > ### Comment · Reviewer_ZW6a · 2023-11-22
> > **Response to the authors**
> >
> > Dear authors,
> >
> > Thank you for answering my questions and to satisfy my curiosity. Thank you for additional experiments that tackle the main contribution of the paper. However, it seems that this approach hardly helps in representation learning as the differences in e.g. classification task are within the statistical error.
> >
> > Therefore, I decide to keep my initial score

---

> > > ### Author Response · Authors · 2023-11-22
> > > **Clarification from the authors**
> > >
> > > Thank you very much for your reply. We agree that improvements on downstream tasks, while consistent across many metrics, are small in magnitude. We had a similar discussion with reviewer nWgS and we now realize that the title of our paper may set wrong expectations. We are open to rephrasing it to “*Investigating* VAE training…” (as allowed by the [updated author guide](https://iclr.cc/Conferences/2024/AuthorGuide)).
> > >
> > > As mentioned in our response to reviewer nWgS, we see the main contribution of our paper not in proposing a better training method but more in *investigating* new ways of addressing the known overfitting problem of VAEs, with a focus on changes to the training data rather than changes to the model architecture or loss function. We do see that the generalization gap shrinks significantly (Figure 2) even though we used an off-the-shelf diffusion model. We clarified in Section 4.1 and in the conclusion that we make the assumptions that (i) cross-model-class distillation requires a generative model that satisfies the two criteria in Table 1, and that (ii) diffusion models satisfy these two criteria. But we hope that our results inspire research that challenges both assumptions, e.g., to develop new generative models that are specifically designed for cross-model-class distillation for VAEs, as this field is much less explored than distillation within discriminative models.

---

### Author Response · Authors · 2023-11-17
**Overall response regarding sample quality and computational cost**

Dear reviewers,

Thank you very much for your insightful comments! Before addressing each review individually, please allow us to quickly address two overall issues:

## (1) Sample quality of the VAEs:
Some reviewers asked about comparisons of sample qualities across the various trained VAEs and the diffusion model. We see little value in using VAEs for data generation as diffusion models are known to produce much better samples. Indeed, comparing FID scores between Appendix C and Table 8 (new Appendix F in updated PDF), we find an FID of 3.9 for the diffusion model and 219 for the VAE (CIFAR-10). Our experiments focus on application domains where VAEs provide an advantage over diffusion models, such as representation learning and reconstruction. Also, as discussed in the introduction, the goal of our method is mainly to fix the overfitting problem of VAEs. As we see in Section 5.3, this problem concerns mostly the encoder, which does not play a role in sample generation.

As diffusion models have recently been receiving so much attention within the generative modeling community, we would like to stress that VAEs provide many applications beyond purely generative modeling tasks that diffusion models do not easily support. These applications are highly relevant in other subfields of machine learning and in other sciences. They include density modeling [(Takahashi et al., 2018)](https://dl.acm.org/doi/abs/10.5555/3304889.3305035), clustering [(Jiang et al., 2017)](https://arxiv.org/abs/1611.05148), nonlinear dimensionality reduction of scientific measurements [(Laloy et al., 2017)](https://www.sciencedirect.com/science/article/pii/S0309170817306243), data compression [(Ballé et al., 2017)](​​https://arxiv.org/abs/1611.01704), and anomaly detection [(Xu et al., 2018)](https://arxiv.org/abs/1802.03903).

## (2) Computational cost:

We agree that our method makes *training* more expensive because one has to train and sample from a diffusion model. However, our method does not change computational cost at test time, which is more important than training time in many applications of VAEs. In addition, training a VAE typically involves trying out various choices of model architectures and hyperparameters, and all these training runs can reuse the same pre-generated samples from the same diffusion model.

But to be specific, Appendix C reports the time for training the diffusion model for CIFAR-10 as 2.5 days, and for sampling of $10 \times |\mathcal D_\text{train}|$ as about 35 minutes on 8 GPUs. Training the VAE (on a single GPU) took 16 hours. We expect this overhead to go down in the future as improving sampling times of diffusion models currently gets a lot of attention (e.g., consistency models, step-aware models, various improvements to the U-Net, other dynamics, ...), and there are also publicly accessible pre-trained diffusion models for many common data sets (e.g., Hugginface).

We look forward to hearing back from you whether this comment clarifies the above two issues. We will address the remaining issues raised by individual reviewers in separate comments.

---

### Author Response · Authors · 2023-11-23
**Summary of updates in the PDF revision**

We would like to thank the reviewers for their time and valuable suggestions on our submission! We believe these suggestions significantly improve the paper. Here, we summarize the major changes in our PDF that resulted from the discussion:

- As there was confusion about our evaluations, we changed the title of Section 5.4 to include “Robustness of Representations” to stress that the analysis in this section (like the analysis in Section 5.3) focuses on the representation learning aspect (encoder) of the VAE, i.e., the application domain where VAEs offer an advantage over diffusion models. We also added additional more direct evaluations of representation learning performance in Appendix F.
- We clarified that we use an *unconditional* diffusion model (footnote on page 4), and that the two criteria in Table 1 (continuous and accurate distribution) are assumptions we make about the diffusion model (Section 4.1) and stated that future work should investigate these assumptions (Section 6).
- We highlighted the gaps shown in Figures 2, 3, 4, and 6 in Section 5 and annotated them with the numerical values. This makes it easier to verify our claim that our proposed method closes the gap.
- We indicated the standard deviation over three random seeds for CIFAR-10 in Figures 2, 3, and 4 (we will do the same for the other data sets once experiments are done). The results remain consistent across random seeds.

Further changes in the appendix:
- Appendix B: fixed an error in Table 3.
- Appendix D: added description on how we picked the hyperparameters.
- Appendix G (new): added an investigation of the distribution of ELBO values on individual data points for CIFAR-10. We did not find any outliers.

We hope that our responses have resolved the reviewers' concerns. We are grateful for the contributions from the reviewers.

---

### Meta-Review · Area_Chair_osW6 · 2023-12-11

**Metareview:**

The paper introduces a novel approach for training Variational Autoencoders (VAEs) using samples from a pre-trained diffusion model to address the overfitting problem in VAE encoders. It presents empirical evidence showing improved generalization performance, amortization gap, and robustness over traditional training methods. The reviewers commend the paper for its clear writing, innovative method, and comprehensive analysis. However, they express concerns about the absence of theoretical underpinnings, the substantial computational costs involved, reliance on the accuracy of the diffusion model, and the method's limited efficacy on complex datasets and in practical scenarios.

**Justification For Why Not Higher Score:**

The reviewers have unanimously identified significant limitations in the paper leading to a consensus that the paper falls below the acceptance threshold.

**Justification For Why Not Lower Score:**

N/A

---

### Decision · Program_Chairs · 2024-01-16

Reject